# A Review of AI-Driven Control Strategies in the Activated Sludge Process with Emphasis on Aeration Control

Celestine Monday [1], Mohamed S. Zaghloul [2], Diwakar Krishnamurthy [3] and Gopal Achari [1,*]

[1] Department of Civil Engineering, University of Calgary, 2500 University Drive NW, Calgary, AB T2N 1N4, Canada; celestine.monday@ucalgary.ca
[2] Department of Civil and Environmental Engineering, United Arab Emirates University, Sheik Khalifa Bin Zayed St., Asharij, Al Ain P.O. Box 15551, United Arab Emirates; m.abdelsamie@uaeu.ac.ae
[3] Department of Electrical and Software Engineering, University of Calgary, 2500 University Drive NW, Calgary, AB T2N 1N4, Canada
* Correspondence: gachari@ucalgary.ca

**Abstract:** Recent concern over energy use in wastewater treatment plants (WWTPs) has spurred research on enhancing efficiency and identifying energy-saving technologies. Treating one cubic meter of wastewater consumes at least 0.18 kWh of electricity. About 50% of the energy consumed during this process is attributed to aeration, which varies based on treatment quality and facility size. To harness energy savings in WWTPs, the transition from traditional controls to artificial intelligence (AI)-based strategies has been observed. Research in this area has demonstrated significant improvements to the efficiency of wastewater treatment. This contribution offers an extensive review of the literature from the past decade. It aims to contribute to the ongoing discourse on improving the efficiency and the sustainability of WWTPs. It covers conventional and advanced control strategies, with a particular emphasis on AI-based control utilizing algorithms such as neural networks and fuzzy logic. The review includes four key areas of wastewater treatment AI research as follows: parameter forecasting, performance analysis, modeling development, and process optimization. It also points out potential disadvantages of using AI controls in WWTPs as well as research gaps such as the limited translation of AI strategies from research to real-world implementation and the challenges associated with implementing AI models outside of simulation environments.

**Keywords:** wastewater treatment; aeration; artificial intelligence (AI); dissolved oxygen (*DO*); control; fuzzy logic

## 1. Introduction

Energy usage is increasing as municipal wastewater treatment plants (WWTPS), also known as water resource recovery facilities, are required to satisfy stricter treatment standards [1,2]. WWTPs require a significant amount of energy for their operation and are often the most significant energy consumers within a municipality [3]. According to data from Italy and Germany, electricity demand for wastewater treatment amounts to around 1% of each country's overall energy usage. In the US, the distribution of water as well as the collection and treatment of wastewater accounts for 4% of the total electricity consumed [4]. The treatment of one cubic meter of wastewater requires about 0.18 to 0.8 kWh of electricity [2,3,5]. In an energy audit of WWTPs across different countries, 388 WWTPs with a collective treatment capacity of about 15.7 million people equivalent (PE) had a corresponding electric energy consumption of $1.72 \times 10^6$ kW h/day [4]. This implies a city with a population of about 500,000 people will consume an average of 54,777 kWh of electricity per day for wastewater treatment, which is enough energy to power a Canadian household for 796.4 days [6].

Most WWTPs utilize aerobic treatment technologies, most commonly the activated sludge process [7]. In aerobic processes, oxygen is supplied to the aeration basin by

pumping air, pure oxygen, or using mechanical aerators. The energy consumption per WWTP unit was evaluated across ten different WWTPs with a representative range of sizes; the percentage of energy consumed by the primary, bioreactor, and clarifier units was 25.08% ± 3.86%, 61.93 ± 8.02%, and 12.69 ± 7.63%, respectively [2]. The bioreactor unit was found to be the most energy-demanding mainly due to aeration. Energy use models for small facilities (serving less than 10,000 people) differ from those for large facilities; depending on the size of the WWTP, aeration consumes about 30% to 70% of the total energy required, with smaller facilities being in the lower range [8,9]. The high energy consumption of the aeration process is because oxygen is not easily transferred from the gaseous phase to the liquid phase for use by microbes in the wastewater [8,10].

The control and optimization of energy consumption in the aeration process have been of significant research interest, and many modifications that have had objectives such as improving the surface contact area of organics, the efficient suspension of materials in the wastewater, energy utilization, and energy recovery have been made. For example, methods including moving bed biofilm reactors (MBBRs) with suspended growth and the moving bed membrane bioreactor (MBMBR) aim to enhance energy efficiency by maximizing surface contact area and enhancing the suspension of organic matter [11,12]. Combined heat and power (CHP) generation using biogas from anaerobic sludge digestion attempts to optimize energy consumption through energy recovery and utilization [13].

Most optimization approaches to the aeration process can be linked to better control of dissolved oxygen (*DO*) concentration. Generally, increasing the air or oxygen flow rate in the aeration basin, which can be achieved with surface aerators or submerged blowers, will significantly improve the *DO*. It is, however, worthy of note that excess airflow can significantly raise energy costs, reduce sludge quality, and adversely affect the overall treatment process [14,15].

There has been a lot of research and development on *DO* control strategies using process control devices. These controllers, collectively known for their diverse functionalities in process management, regulate aeration device components, such as blowers and valves, based on incoming process signals. There has also been an increase in the sophistication of controllers from conventional on–off and proportional integral derivative (PID) controllers to advanced control strategies like adaptive fuzzy control, adaptive PID, and artificial intelligence-based controls.

Conducting a bibliographic search in the Scopus database within the engineering and environmental science domain using the keyword "Artificial Intelligence" showed an exponential increase in published works on this subject. A similar search using "Artificial AND Intelligence AND Wastewater" showed a parallel exponential increase, indicating a growing interest in applying Artificial Intelligence (AI) to wastewater-related contexts.

The inception of AI as a field of study traces back to a foundational assumption posited by researchers at Dartmouth College in 1956. They envisioned the possibility of describing every facet of learning and intelligence, paving the way for machines to emulate these characteristics [16]. Today, AI refers to computerized systems capable of performing tasks traditionally requiring human intelligence, covering perception, reasoning, learning, and decision making. Within AI, machine learning (ML) is a key component, employing statistical and computational algorithms to empower computers to learn from data and enhance their task performance over time. ML branches into the four primary classes of supervised, unsupervised, semi-supervised, and reinforcement learning. Supervised learning trains models using labeled data for predictive tasks, while unsupervised learning detects patterns in unlabeled data. Semi-supervised learning merges these approaches, and reinforcement learning trains models to maximize rewards within an environment.

The integration of numerous ML algorithms, such as artificial neural networks, genetic algorithms, and fuzzy logic is gaining traction within WWTPs [17]. Diverse publications show that AI and ML in WWTPs largely serve purposes like process modeling, control, performance analysis, forecasting, and optimization.

This paper is based on literature published within the past decade and attempts to identify the following points: (i) factors that influence oxygen transfer from the gaseous to liquid phase, (ii) systems required for the implementation of aeration control, (iii) different aeration control strategies that have been implemented both in simulation and pilot scales, and (iv) advanced aeration control strategies, particularly AI control strategies. The paper also identifies research gaps such as the implementation challenges of AI strategies in operational plants and the trade-off between model complexity and performance. Furthermore, it highlights recommendations on generally accepted principles and good practices that should be considered to bridge the gaps.

The paper's structure is as follows: Section 2 explores oxygen demand and the factors influencing its transfer efficiency. Following that, Section 3 examines various components crucial for WWTP control. This includes an overview of control platforms, structures, models, and algorithms. Furthermore, Section 3 extensively explores control strategies, particularly focusing on AI in control strategies. This section also delves into real-world WWTP implementations of AI control while shedding light on the potential disadvantages associated with such implementations. Sections 4 and 5, respectively, address the existing research gaps identified throughout the paper and the conclusions drawn from the study's findings.

This study's contribution to research lies in its comprehensive exploration and analysis of the integration of AI within WWTP control strategies. By highlighting the nuances, challenges, and opportunities associated with AI-driven control strategies, this study provides valuable insights for practitioners, researchers, and policymakers in the field of wastewater treatment. Furthermore, the review aims to contribute to the ongoing discourse on enhancing the efficiency, reliability, and sustainability of WWTPs through the integration of advanced AI-based control systems, paving the way for informed decision making and future advancements in this critical domain.

## 2. Oxygen Demand in the Biological Treatment Process

Aeration is essential for nitrification and breaking down organic substances in biological wastewater treatment. However, the aeration process itself is a complex, nonlinear system involving more than blowers alone as well as an intricate air delivery setup, as represented in Figure 1. The most utilized modeling framework for both the aeration system and the entire biological treatment process is covered by the activated sludge model (ASM) [18]. These models comprise an extensive number of state variables as well as kinetic and stoichiometric parameters. In most cases, a simplified approach is used to estimate the dynamic oxygen demand. This simplified model operates under certain assumptions; it assumes a single type of microorganism and substrate, focusing solely on organic matter removal. Additionally, it presumes continuous stirring in the aeration tank, no reactions in the settler, and exclusively recycles activated sludge to provide it back to the aerated bioreactor. Along the recycling path, this model disregards the concentrations of oxygen and the substrate. It also assumes that the output flow from the aerated bioreactor equals the combined output flow from the settler and the recycled activated sludge flow [19]. The amount of oxygen required by the microorganisms during wastewater treatment varies and is given using the dynamic *DO* mass balance in the aerated bioreactor in Equation (1) [20]:

$$\frac{dS_o(t)}{dt} = \frac{Q_{in}(t)\, S_{oin}(t) - Q_{out}(t)\, S_o(t)}{V(t)} + kLa\,(Q_{air}(t))(S_{o,sat} - S_o(t)) - \frac{S_o(t)}{K_o + S_o(t)} Rr(t) \qquad (1)$$

where $Q_{in}$ [m$^3$/h], $Q_{out}$ [m$^3$/h], $Q_{air}$ [m$^3$/h], $S_{oin}$ [g.O$_2$/m$^3$], $S_o$ [g.O$_2$/m$^3$], $Rr$ [g/m$^3$h], $V$ [m$^3$], $K_o$ [g/m$^3$], $S_{o,sat}$ [g.O$_2$/m$^3$] represent the waste inflow into the aerated bioreactor, the waste outflow out of the aerated bioreactor, the airflow, the influent *DO* concentration, the *DO* concentration in the aerated bioreactor, the respiration, the volume of the bioreactor, the Monod constant and the *DO* concentration saturation limit respectively.

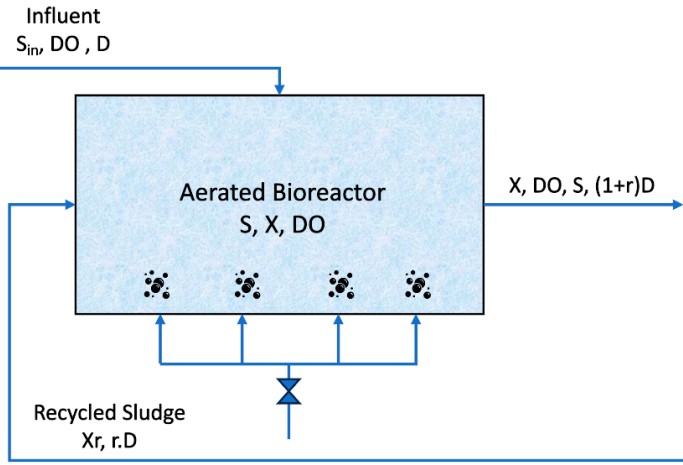

**Figure 1.** Simplified process of an aerated bioreactor. $S_{in}$ = Influent substrate concentration, DO = Dissolved oxygen concentration, D = Dilution rate, S = Substrate concentration, X = Biomass concentration, r = Ratio of recycled sludge flow, $X_r$ = Recycle biomass concertation.

It is also worthy of note that oxygen does not transfer easily from a gaseous to a liquid phase. The parameters that define oxygen transfer from the gaseous to the liquid phase are derived from Fick's first law of diffusion, which relates the flux of a substance to its concentration gradient:

$$\frac{\partial M}{\partial t} = -DA\frac{\partial C}{\partial x_f} \tag{2}$$

where:

$\frac{\partial M}{\partial t}$ = rate of oxygen transfer (g.$O_2$/h);

D = diffusion coefficient ($m^2$/h);

A = interfacial contact area between the gaseous and liquid phase ($m^2$);

$\frac{\partial C}{\partial x_f}$ = oxygen concentration gradient (g/$m^4$).

Following some derivations, Equation (2) becomes the oxygen transfer equation as follows [15,21]:

$$\frac{dC}{dt} = K_{La}(C_s - C) = K_{La}(C_s - DO) \tag{3}$$

where:

$Cs$ = dissolved oxygen concentration at the gas–liquid film interface (g.$O_2$/$m^3$);

C = dissolved oxygen concentration in bulk liquid volume (g.$O_2$/$m^3$);

$K_{La}$ = transfer coefficient (1/h).

Equation (3) forms the basis of modeling the oxygen transfer and is referred to as the oxygen transfer rate (OTR) equation. Numerous studies have shown that other factors that are not part of the OTR equation can also have effects on the OTR. These factors, such as surface-active agents and salinity, are discussed below.

### 2.1. Surface-Active Agents

Surface active agents (SAAs), or surfactants, have been identified as major contaminants in wastewater [22]. The unique properties of SAAs have made them widely used in various industries, such as the food, medical, and chemical industries, which explain their continuous presence in the wastewater treatment (WWT) influent streams [23]. They can plug subsurface aeration devices that inhibit oxygen transfer to aerobic microorganisms, and they can have antibacterial properties depending on their compositions [23,24].

It has been observed that surfactants indirectly reduce the oxygen transfer coefficient ($K_{LA}$) of fine bubble aerators by reducing the bubble surface area, though they improve the $K_{LA}$ of surface aerators and coarse bubble diffusers [15,21]. This variation in the impact of

SAAs on fine bubble and surface aerators has been attributed to the turbulence associated with oxygen transfer using the different aerators and the velocities at which the air bubble travels through the wastewater.

### 2.2. Wastewater Salinity

Regional freshwater scarcity in many areas of the world has led to the use of saline water as an alternative to freshwater, especially for industrial purposes [22]. The byproduct of the desalination process is brine; when it is discharged into the wastewater stream, the salt concentration in the wastewater ranges from 15 to 45 g/L [25].

Different researchers have conducted studies on saline wastewater, and it has been observed that an increase in the concentrations of salinity decreases the OTR and further affects the activated sludge microbial consortia. In a study on the effect of different salinity concentrations (0 to 3% $w/w$) on the performance of microbial consortia in a sequential batch reactor (SBR), an optimal biological nutrient removal performance was observed when no salinity was present [26]. The performance decreased by 5% at a salinity concentration of 2% ($w/w$). In another study with synthetic wastewater samples containing NaCl, changes in microbial consortia were observed as the NaCl concentration increased from 5 to 20 g/L, but there was no observed change in the nitrification performance [27]. It was, however, noted that nitrification was affected only when NaCl salt concentrations increased rapidly. This was validated in a different study using a fixed-bed biofilm CANON bioreactor and wastewater samples containing NaCl concentrations ranging from 0 to 45 g/L [28]. Inhibition of nitrification was observed with higher salt concentration due to the rapid decrease in ammonia-oxidizing bacteria (AOB) and nitrite-oxidizing bacteria (NOB).

Salinity also influences the sludge volume index (SVI). A gradual increase in salinity reduces the SVI because filamentous bacteria do not thrive in environments with high salinity [27].

These factors may not be under the direct control of the WWTP operator but could be altered to improve treatment efficiency. Other factors in works of the literature include mixed liquor concentrations, solid retention time, and diffuser-related factors such as depth, fouling, scaling, and temperature.

## 3. Aeration Control

Over the years, the WWT process has undergone numerous advancements using increasingly complex designs to meet treatment standards and effluent discharge regulations. The WWT influents not only have varying flows but also have varying concentrations. Human operators can apply experiential knowledge and rules of thumb to implicitly control the WWTP, but the process is nonlinear and has a multi-variable timescale, thus requiring more than the experiential knowledge of the operator alone [29]. Control mechanisms are tools that are preferably used to maintain a high-level performance of the activated sludge process (ASP) amidst the system's variabilities and disturbances. The two main control elements comprise the air supply system's capacity for varying airflows over intervals and the sensors that transmit signals to prompt control actions [10]. Most aeration systems are controlled based on the *DO* levels within the aeration basin and can save 25% to 40% of energy costs [30,31].

### 3.1. Control Implementation Platforms

All sizes of WWTPs require a control interface to monitor plant operations and sensor signals while providing some level of control. To implement WWTP control, three basic control platforms are extensively employed. Supervisory control and data acquisition (SCADA), distributed control systems (DCSs), and programmable logic controllers (PLCs) constitute the three types of control interfaces. Although a PLC can be used on its own, SCADA and DCSs are always integrated into PLCs.

PLCs are originally designed to meet the control requirement of a single piece or discrete component of a process, while DCSs are designed to provide local centralized data

collection from multiple pieces of connected process components [32]. SCADA is widely used in WWTPs as it has the advantage of notifying an operator of possible malfunctions, remotely turning process components on or off, displaying and logging real-time data, and providing a plant-wide view [32,33].

Unlike a DCS, SCADA can be accessed at any time by the operator without the constraint of space or distance; however, it has a relatively slower communication time as SCADA systems are online while DCSs are onsite [34]. The preference for one control platform over another is based on the operator's choice, economic factor, or platform compatibility with the process equipment.

### 3.2. Conventional Control Strategy

3.2.1. The On–Off Control

Being also referred to as a traditional control strategy, on–off controls are known for their simplicity in turning the process of a manipulated variable off or on when an output signal is greater or less than a control setpoint or a dead band is received. For instance, a *DO* controller with a setpoint of 2.0 mg/L and a dead band of ±0.5 is illustrated in Figure 2. Despite its simplicity, it is one of the least preferred control algorithms for WWTPs the as on–off switching of the inputs can cause process disturbances and huge stresses [20].

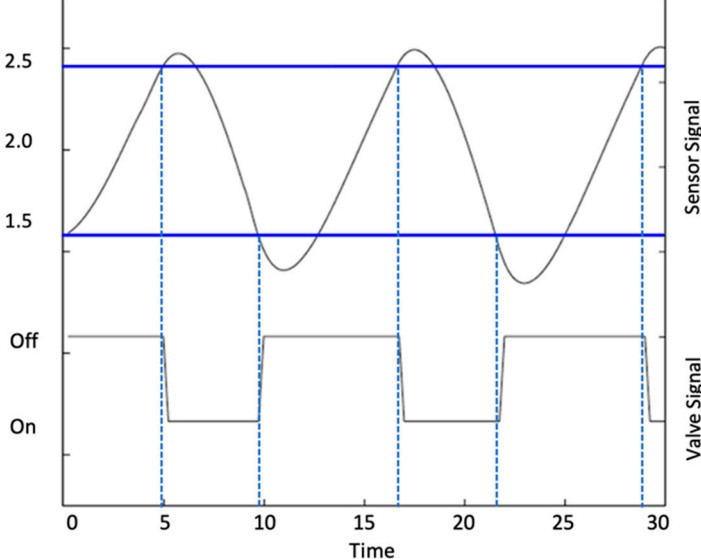

**Figure 2.** On–off control strategy showing sensor and corresponding valve signal.

3.2.2. PID Control

Although the WWT is a nonlinear process, linear control strategies such as the PID are the most deployed in full-scale WWTP applications [35]. As seen in Equation (4) and Figure 3, the PID combines proportional (P), integral (I), and derivative (D) control actions that are applied proportionally to the error between the desired and measured feedback signals. PID controllers are commonly used in the variants of the P and PI control actions [36]. Unlike the on–off control strategy, the PID eliminates the oscillations, thus keeping the process at the set point.

$$u = u_o + K_P \cdot e + K_I \int e \cdot dt + K_D \frac{de}{dt} \qquad (4)$$

where:

$e = (u - u_o)$ is the error;
$K_P$ = proportional gain;
$K_I$ = integral gain;
$K_D$ = derivative gain.

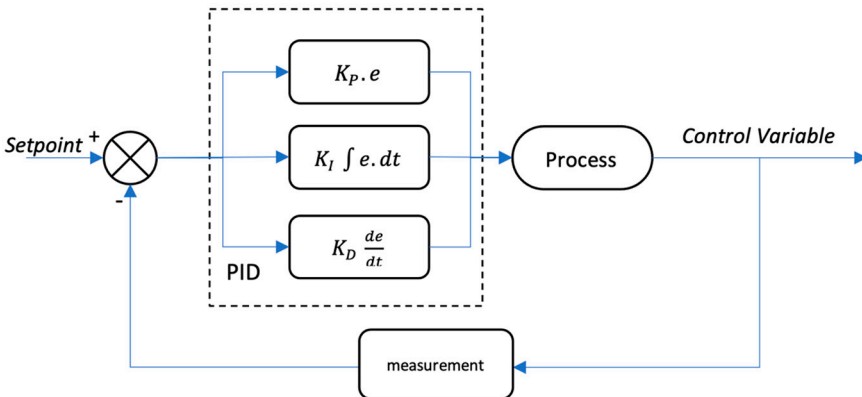

**Figure 3.** PID control structure.

The PID control strategy is reliable when the system is stable, but proper tuning is required to maintain its reliability when there are disturbances [35,36].

*3.3. Control Structures*

3.3.1. *DO* Cascade Control

Control structures organize the controlled and manipulated variables within a control loop, making a process more responsive to disturbances and less susceptible to downtime. As illustrated in Figure 4, a cascade control structure is made up of multiple loops and controllers in which one or more controllers calculate a set point for others. The simplest form of a cascade control structure comprises two controllers and control loops that monitor two measurement signals to control one primary variable [36].

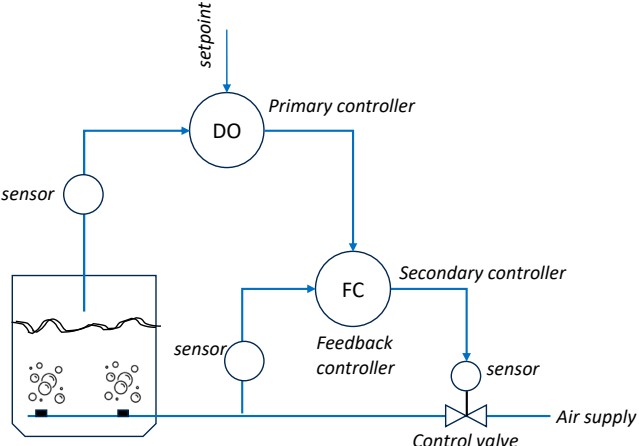

**Figure 4.** A simple cascade control structure.

A typical application of a cascade structure is highlighted below in the ammonia-based aeration control (ABAC) section, where an ammonium reference and concentration values are used by a primary ABAC controller to compute a set point for a *DO* controller. Another application is in the *DO* and air flowrate cascade controller, where the *DO* controller determines the setpoints for the air flowrate controller to manipulate valve positions.

3.3.2. Ammonia-Based Aeration Control (ABAC)

Removal of ammonia and nitrogen from wastewater is carried out through nitrification and denitrification, respectively. Theoretically, complete nitrification is an oxygen-intensive reaction in which 4.57 g of oxygen are required.

ABAC is a cascade-type control structure, as shown in Figure 5. It prevents the complete nitrification of ammonia while keeping effluent setpoint values below the discharge concentration that is permitted by municipal regulations [31,37]. ABAC has proved that

nitrifying enough $NH_4^+$ to meet regulatory standards saves energy used for aeration [38]. Based on the direction of signal flow, ABAC can be categorized as a feedforward or a feedback loop. Feedback-controlled loops measure the control variable, which is then used as an input for the controller while feedforward-controlled loops measure process disturbances and use a predictive model to map out the behavior of the controlled system to take a control action [31].

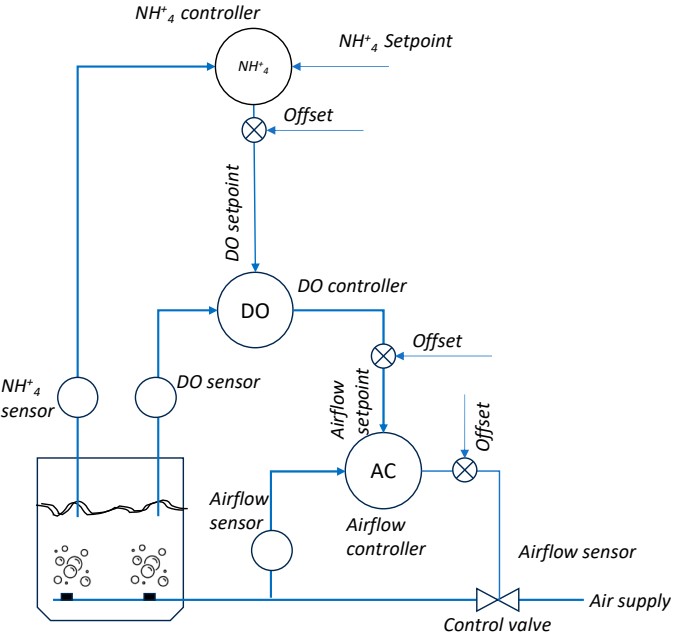

**Figure 5.** Ammonia-based aeration control (ABAC) structure.

### 3.4. Artificial Intelligence (AI) Control Strategy—Advanced Control

Advanced control strategies refer to a wide range of techniques and technologies used within process control systems, usually together with basic or conventional process controls for optimization, accounting for nonlinearities and other system constraints. Advanced aeration control strategies are seldom used in field applications because of their complexities and industrial viability [37]. In this section, the advanced controls of focus are AI control strategies. As shown in Figures 6 and 7, there is an observed increase in interest in the application of AI in environmental science and wastewater-related contexts.

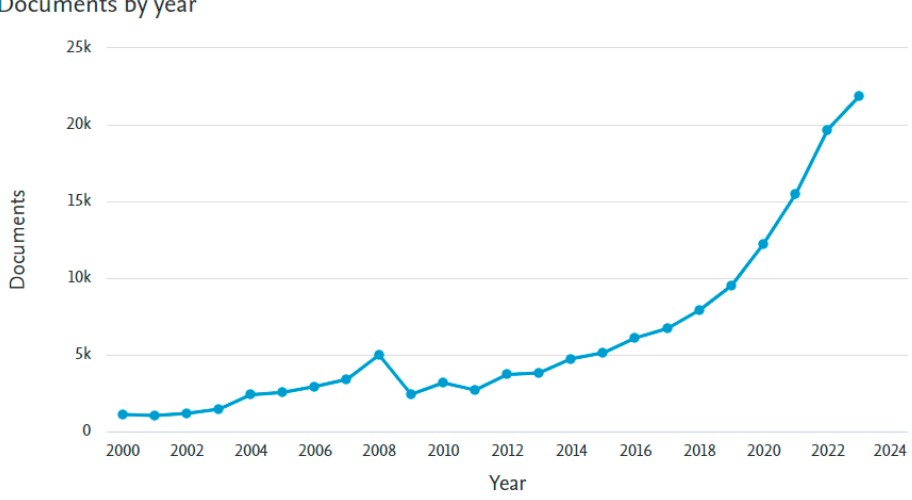

**Figure 6.** Bibliographic search with the keyword "Artificial AND Intelligence" across engineering and environmental science domains.

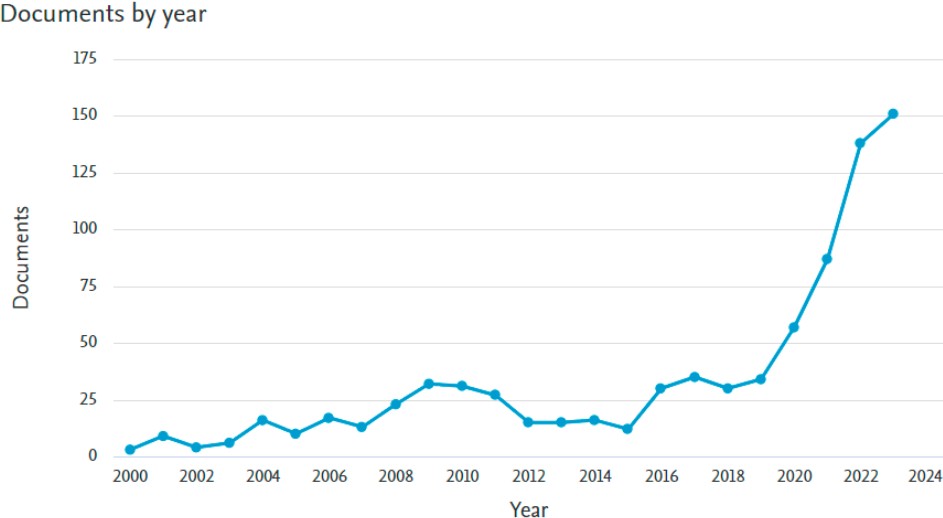

**Figure 7.** Bibliographic search with the keyword "Artificial AND Intelligence AND Wastewater" across all domains.

AI is a branch of computer science that integrates a broad range of techniques to imitate human intelligence such as natural language comprehension, object identification, decision making, and learning from experience. Machine learning (ML), which is often conflicted with AI, is an AI technique that utilizes algorithms to enable computers to learn from data and improve task performance. While there are numerous AI and ML algorithms, the most documented in works of the literature are artificial neural networks (ANNs), genetic algorithms (GAs), and fuzzy logic (FL) [17,39]. As shown in Figure 8, AI technologies applied in WWTP research have mainly been used to study parameter forecasting, performance analysis, decision support, modeling development, process optimization, and the development of smart control systems.

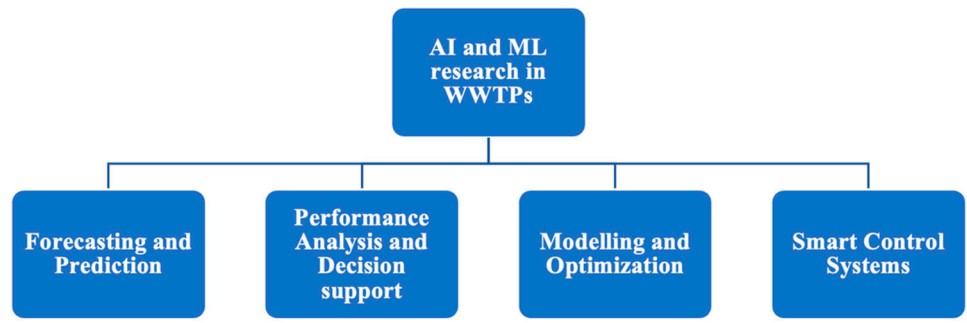

**Figure 8.** Areas of AI and ML research in WWTPs.

AI technologies used for forecasting and prediction play a role in supporting plant operators with advanced information about impending events. This is particularly valuable in predictive maintenance and anticipating time-series events, such as an expected spike in chemical oxygen demand (COD) levels. Some algorithms in this domain include Autoregressive Integrated Moving Average (ARIMA) and its variant, Auto-regressive Moving Average (ARMA). Moving to performance analysis and decision support, AI technologies here, serve as a secondary layer for validating process operations and efficiency. This may involve the utilization of soft sensors, enhancing the robustness of assessments. Furthermore, AI tools for modeling and optimization within WWTPs function to delineate specific operations in a replicable manner, allowing for the observation of operational policies with various modifications. The primary aim is process optimization without necessitating these changes within the actual WWTP itself. Lastly, smart control systems leverage algorithms to implement aspects of prediction, forecasting, decision support analy-

sis, and optimization. These systems are typically commercialized as Software-as-a-Service (SaaS) applications, providing a versatile and scalable approach to implementing AI-driven functionalities within wastewater treatment processes. Some frequently used algorithms applied in WWTPs, and their use cases are discussed in more detail below.

### 3.4.1. Fuzzy Logic Control (FLC) Strategy

FLC is a nonlinear control algorithm that belongs to a family of rule-based controllers; unlike conventional or Boolean logic (e.g., on–off) controllers, it uses successive intervals between 0 (off) and 1 (on). FLCs integrate linguistic information from human experts and process knowledge to make up a linguistic rule base for the inference mechanism of the controller. As shown in Figure 9, the FLC has the four main components the fuzzy inference system (FIS), fuzzifier, de-fuzzifier, and fuzzy rules [39,40]. From a high-level perspective, FLCs use membership functions to fuzzify crisp input signals, which are a set of linguistic rules for inference mechanisms, and a de-fuzzifier to transform control outputs into crisp forms. FLC applications have been found in plant-wide biological nutrient removal, greenhouse gas emission control, and in $N_2O$ emission reductions of up to 35% [41,42]. When FLC was compared with PID controllers in varying control structures, FLC was observed to improve process stability and energy savings [35]. Benchmark simulation model 1 (BSM1) mirrors the operations of a real wastewater treatment plant in a simulated setting. It utilizes a standard control approach with proportional integral (PI) controllers and is widely employed in simulating various control strategies. Using FLC in BSM1 with MATLAB/Simulink in which *DO* was the control variable also recorded significant performances, such as a 15.57% increase in effluent quality [43,44]. FLC has some variants with high control performance that are usually applied to manage conditions of unforeseen disturbances on the system or when a model of the controlled process is unavailable. These variants include the adaptive fuzzy-neural network (AFNN), the fuzzy-neural network (FNN), supervisory committee fuzzy logic (SCFL), and the adaptive neuro-fuzzy inference system (ANFIS) [40,45–47]. Of these variants, the ANFIS does not require an expert or operator to develop a rule base. Its ANN component specifies membership functions and infers rules from training on process data [40]. The ANFIS and model predictive controller (MPC) were used on a large-scale WWTP of about 500,000 people equivalent [39]. The ANFIS–MPC configuration recorded obtained better effluent quality and energy savings when compared with the PIDs previously used. In a study comparing ANFIS and generalized linear model (GLM) regression, ANFIS models provided better predictions of the studied effluent variables [48]. FLC is one of the most popular control strategies and among the few that have been validated through full-scale application [49]. Table 1 provides a list of some fuzzy logic AI techniques used for WWTP control.

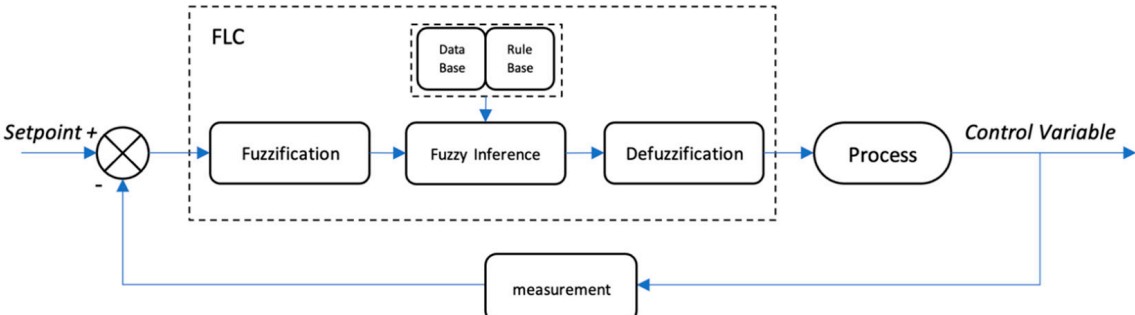

**Figure 9.** FLC control structure.

**Table 1.** Fuzzy logic AI techniques used for WWTP control.

| Model Type | Application | Controlled Variable(s) | Metric of Evaluation (Compared with Default Controller) | Reference |
|---|---|---|---|---|
| MPC based on neuro-fuzzy control | Field—500,000 PE | *DO* Recycle flow (Qr) | 16% energy saving 8.1% reduction in effluent N total | [39] |
| FLC in cascade with PI | Simulation—BSM2G | Recycle flow rate (Q) External flow rate Dissolved oxygen Setpoint (DOsp) | 1.73% on EQI (effluent quality index) 17.20% on OCI (operating cost index) 8.60% of total $CO_2$ | [41] |
| FLC | Simulation—BSM2 Field—3500 PE | Effluent ammonia concentration | +7–8% on EQI (simulation) +13% on energy saving (simulation) +40–50% energy saving (field) | [35] |
| FLC in cascade with PI | Simulation—BSM1 | Effluent ammonia concentration Recycle flow (Qr) | +15.57 to 20.3% effluent quality | [44] |
| FLC | Simulation—BSM1 | Dissolved oxygen (*DO*) | Faster rejection of disturbance to maintain a set point compared with the PID | [43] |
| FLC | Simulation—BSM2N | The ratio of nitrate produced by NOB and the ammonium consumed. by AOB | 35% reduction in $N_2O$ emissions | [42] |
| AFNN + optimization algorithm | Simulation—BSM1 | *DO* setpoint (DOsp) and $NO_2$ setpoint | 7% aeration energy 8% pumping energy | [45] |
| Type-2 fuzzy broad learning controller | Simulation—BSM1 | *DO* and $NO_3$-N | Compared with FNN: 70% faster computational time 90% + less integral square error | [50] |
| Fuzzy-based predictive controller | Simulation—BSM1 | *DO* | Faster rejection of disturbance to maintain set point compared with the PID | [51] |
| Oxy fuzzy logic | Field—75,000 PE | Effluent ammonia concentration ($NH_3$) | 13% reduction in annual energy consumption | [52] |
| Fuzzy-based MPC | Simulation—BSM1 | *DO* | Faster rejection of disturbance to maintain set point compared with the PID | [53] |
| FLC | Simulation—BSM2 | Recirculation flow rate (Qa) $NH_3$-H | 2.25% to 57.94% reduction in Ntot violations 55.22% to 79.69% reduction in $NH_3$-N limit violations 0.84% to 38.06% reduction in the cost of pumping energy | [54] |
| Fuzzy-neural network controller (FNNC) + multi-objective optimal control (MOOC) | Simulation—BSM1 | *DO* and $NO_3$ | Energy consumption (EC) reduced by 1.6% in dry, 1.15% in rain, and 2.17% in storm conditions | [55] |
| Cooperative fuzzy-neural control (CFNC) | Simulation—BSM1 | *DO* and $NO_3$-N | 0.0021 *DO*-integral of absolute error (*DO*-IAE) 0.2357 *DO*-integrated square differential error (ISDE) 0.0049—$NO_3$-N—IAE 0.4587—$NO_3$-N—ISDE | [56] |
| Cooperative fuzzy-neural control (CFNC) | Field—16,000 m³/d | *DO* and $NO_3$-N | 0.0084 *DO*-IAE 0.3677 *DO*-ISD 0.0143—$NO_3$-N—IAE 0.4987—$NO_3$-N—ISDE | [56] |

### 3.4.2. Artificial Neural Network (ANN) Control Strategy

ANNs are typically stand-alone AI technologies, but there are several types of ANNs that include the radial bias function (RBF), wavelength neural network (WNN), multilayer perception (MLP), and self-organizing map (SOM) [17]. Structurally, all neural networks are made up of input, hidden, and output layers, interconnected through weighted synaptic

connections using groups of nodes to infer approximate functions between input and output variables. ANN control strategies use the gradient descent method and are trained to recognize patterns from a dataset. The inference drawn from the training dataset can be applied to the experimental designing and solving of multivariant nonlinear problems. ANN control strategies have been observed to have some limitations, such as high training cost, slow convergence, and the possibility of local minimum entrapment, when the network has enormous input disturbances [57–59]. Throughout the literature, ANNs are often used together in an ensemble with other models such as the GA, FL, MPC, and internal model control (IMC) in control strategies [60–63].

In BSM1, *DO* was controlled using IMC and a multilayer perceptron neural network [62,63]. These methods were employed to regulate *DO* levels by adjusting the oxygen transfer coefficient ($K_{La}$). The comparison of performance with the default PI of BSM1 indicated at least a 42% decrease in the integral squared error (ISE) amongst other metrics of performance. Similarly, within the BSM1 framework, the control of *DO* and nitrate levels was accomplished by manipulating the oxygen transfer coefficient and internal recycling rates, respectively [55]. This was achieved through the development of a multi-objective optimal control system utilizing a fuzzy-neural network controller. Notably, this approach demonstrated improved performance not only compared with the default PI controller but also with other optimal controllers documented in the existing literature.

To overcome the challenge of local minima entrapment, which was a limitation of the traditional neural network as mentioned earlier, an online sequential extreme learning machine (OS-ELM) was introduced. An OS-ELM is a single-layered feedforward neural network designed to address this issue [58]. When implemented in the BSM1, the $K_{La}$ and *DO* in the current time step were controlled to yield *DO* for the next time step, the OS-ELM achieved impressive results. It recorded integral of absolute error (IAE) values of 0.0475 and integral of squared error (ISE) values of 0.00069 both in the dry weather and rainy weather conditions of the BSM 1.

Utilizing artificial neural networks (ANNs) in an ensemble allows for the exploration of diverse ANN-based control strategies, enhancing the adaptability and effectiveness of the overall system. Table 2 provides a list of some ANN techniques used for WWTP control.

**Table 2.** Artificial neural network (ANN) AI techniques used for WWTP control.

| Model Type | Type of Test | Controlled Variable(s) | Metric of Evaluation (Compared to Default Controller) | Reference |
|---|---|---|---|---|
| ANN-based internal model control (IMC) | Simulation—BSM1 | *DO* | 16% IAE<br>53% ISE | [63] |
| ANN-based IMC | Simulation—BSM1 | *DO* | 21.25% IAE<br>54.64% ISE | [62] |
| Fuzzy-neural network controller (FNNC) + multi-objective optimal control (MOOC) | Simulation—BSM1 | *DO* and $NO_3$ | Energy consumption (EC) reduced by 1.6% in dry, 1.15% in rain, and 2.17% in storm conditions | [55] |
| Adaptive control based on online sequential extreme learning machine (OS-ELM) neural network | Simulation—BSM1 | *DO* | Dry weather:<br>IAE—0.0475<br>ISE—0.00069<br>Rain Weather:<br>IAE—0.0375<br>ISE—0.00067 | [58] |

### 3.4.3. Genetic Algorithm (GA) Control Strategy

GA is an optimization algorithm inspired by Darwin's theory of evolution; it is also one of the most refereed examples of evolutionary computation. One of the peculiarities of GA optimization is its implementation with evolutionary-biology terms. For example, selection, crossover, and mutation are operations applied to the population of chromosomes (viable

solutions) to find an optimal solution (off-spring) [64]. GA can search through complex spaces independently of space dimensionality at a high performance and can be a soft calculation tool for systems with high nonlinearities such as WWTPs [65].

It is important to note that GA is an optimization algorithm and is typically integrated into other AI control strategies mainly to reduce the error margin of the predictive AI model itself and to improve operational efficiency by optimizing the controlled variable. An instance where GA control was applied to improve operational efficiency was in the hierarchical control structure proposed for biological nutrient removal operations [65]. Here, iterative learning and PIs were used in a lower-level control of *DO* by manipulating the $K_La$ while a GA was used at a higher level to determine the *DO* setpoint at optimal OCI and EQI values. This higher-level control design based on a GA for WWTPSs recorded a noticeably reduced OCI, which is a direct measure of oxygen consumption [65]. Research in which a GA is used to reduce the error margin of predictive AI models used for the control of WWTPs is more common in the literature. Some of these instances include the FLC-based GA control of *DO* and a GA evolving fuzzy wavelet neural network (FWNN) control of *DO* [66,67]. Where a FLC-based GA was used to control *DO*, the membership function of the FLC was optimized by the GA. Comparisons with the default PI controller showed that the FLC-based GA controller had a 22.12% and 9.63% reduction in integral squared errors (ISEs) and integral absolute errors (IAEs), respectively [66]. For the GA evolving FWNN, the GA was used to adjust the center and width parameters of the Gaussian function, dilation, and translation parameters of the wavelet functions as well as the weight of the wavelet networks [67].

### 3.4.4. AI-Driven Model Predictive Control (MPC)

An MPC is a model-based controller used for advanced control strategies, and their models could be linear or nonlinear. AI-driven MPCs are nonlinear and use AI models such as FL and ANN MPCs have the characteristic of handling multiple inputs and outputs (MIMO) with constraints that could be varied as hard or soft. The primary feature of an MPC is its use of an optimization algorithm and a plant model to solve the control problem and predict plant behavior, respectively, over a future horizon [61,68]. At each control time step ($k$), both over a prediction horizon ($p$) and within a control horizon ($m$), MPCs compute a series of control moves with predictions over p as:

$$\hat{y}(k+1|k), \hat{y}(k+2|k), \ldots, \hat{y}(k+p|k) \tag{5}$$

and optimizations over $m$:

$$\Delta u(k), \Delta u(k+1), \ldots, \Delta u(k+m-1). \tag{6}$$

For a linear industrial process with linear constraint and cost functions, a linear time-invariant MPC, such as an adaptive MPC or gain-scheduling MPC, can be implemented. A WWTP is a nonlinear system, but its process can be approximated in the vicinity of a working point by a discrete-time state-space model [69]. Nonlinear MPCs (NMPCs) have gained popularity since the 1990s; they are more appropriate for controlling highly nonlinear systems like WWTPs [36]. AI-driven MPCs are NMPCs that use AI models to predict plant behavior. Certain AI-driven MPC configurations include the fuzzy-supervised NMPC, MPC + FLC configuration that is implemented for the removal of effluent violations in WWTPS, the advanced decision control system with MPC+ Feedforward (FF), FLC, and an ANN MPC + FF [61,69,70]. Moreover, a process control scheme was developed with an MPC in which the set points for *DO* and the recycle flow ($Q_r$) were forecasted with the AI engine of the ANFIS [39].

While MPCs provide the benefit a reduction of over 25% in power consumption as well as an increase in plant efficiency, it is a computationally intensive process when computations are conducted over smaller time steps ($k$) as well as with longer prediction ($p$) and control ($m$) horizons [68]. To manage the high computational burden, viable

recommendations within a simulation space are event-triggered [71,72]. Table 3 provides a list of some MPC techniques used for WWTP control.

**Table 3.** Model-predictive control (MPC) AI techniques used for WWTP control.

| Model Type | Type of Test | Controlled Variable(s) | Metric of Evaluation (Compared with Default Controller) | Reference |
|---|---|---|---|---|
| Nonlinear multi-objective model-predictive control (NMMPC) | Simulation—BSM1 | $DO$ and $NO_3$ | 3.2% to 9.1% aeration energy | [68] |
| MPC + FF (feedforward) and FL | Simulation—BSM1 | $DO$ and $NO_2$ | 3.9% on OCI 5% on EQI | [69] |
| Hierarchical structured MPC + FF, FL, and ANN MPC + FF | Simulation—BSM2 | $DO$ and $NH_3$-N | 2.62% to 37.09% OCI 3.41% to 12.6% EQI | [61] |
| Fuzzy-supervised NMPC | Benchmark Simulation with ASM2D | TN and TP | 18% reduction in plant operating cost | [70] |
| Event-triggered MPC (ETMPC) | Simulation | $DO$ and $NO_3$ | 60% computation reduction and 0.1 improvements for the integral of the squared error (ISE) | [72] |
| Event-triggered NMPC (ETNMPC) | Simulation—BSM1 | $DO$ and $NO_3$ | 50% computation reduction | [71] |
| Fuzzy-based MPC | Simulation—BSM1 | $DO$ | Faster rejection of disturbance to maintain set point compared with the PID | [53] |
| NMPC | Simulation—BSM1 | $DO$ and $NO_3$ | 20% reduction in operation costs | [73] |
| MPC + genetic algorithm (GA) | Field—4000 PE | $DO$ | 50% reduction in the relative amount of aeration used | [74] |

### 3.4.5. Machine Learning and Data Mining (ML-DM) Control and Optimization

As highlighted in an earlier section, AI technologies have broadly been grouped into ANN, FL, and GA. These three technologies have been referred to as typical stand-alone AI technologies [17]. However, there are other AI methodologies applied to WWTP controls. Some of these methodologies, such as random forest (RF) and K-nearest neighbor (KNN), have been collectively referred to as machine learning (ML) techniques in some sections of the literature and data mining (DM) techniques in others. Although ML and DM differ in terms of purpose, learning functionality, human interaction, and their functional design to self-improve, they use similar algorithms, such as classification and regression. While DM is a computer-assisted process of revealing non-trivial patterns and establishing concise relationships within an enormous data set, ML is a collection of methods through which computers automate data-driven models and discover non-trivial patterns in data without being programed for specific problems [75,76].

ML and DM are mostly applied for prediction and offline optimization. The offline prediction and optimization models can subsequently be used to develop a control strategy. In this section, some ML-DM techniques that have been applied to the prediction, control, and optimization of the WWTP ASP are collectively discussed as ML-DM control strategies. A study that aimed to improve the aeration process of Detroit's water and sewerage used an offline modeling methodology with different ML-DM methods such as the multi-adaptive regression spline (MARS), RF, and KNN [77]. The model obtained from MARS was comparatively selected for WWTP optimization. The results obtained yielded an airflow reduction of greater than 31%. MARS was also used comparatively with reinforcement learning (RL) and a constrained Markov decision process (CMDP) for the WWTP of Lleida, Spain [78]. In that study, RL had some calibration limitations, while MARS had a long runtime of four hours compared with the CMDP, which ran successfully in eight seconds. Building on this comparative base, a CMDP was used for the Lleida WWTP pilot test with

results of 13.5%, 14%, and 17% reductions in the plant's electricity consumption, chemicals needed for phosphorus removal, and sludge production, respectively.

RL is an advanced ML-DM methodology that differs from the conventional ML-DM methodologies as it does not require training datasets. RL utilizes an agent to learn and conduct better optimizations within an environment by interacting directly with the environment. Some RL applications in WWTP systems include control tracking and the cost decrease in N-ammonia removal [79,80].

A single ML-DM methodology can be used to develop models that may or may not substantially characterize the nonlinearities of a WWTP process. However, a robust characterization can be achieved by using different models at different points in the ML-DM pipeline where an RF model was developed and validated using a deep neural network (DNN) [81]. In the same study, variable importance measure (VIM) analyses were used to determine the feature importance of the total suspended solids in the effluent (TSSe), while a partial dependence plot (PDP) analysis was used to explain the effect of the feature from the VIM on the TSSe.

Alternatively, a more robust ML-DM model can be achieved by using an ensemble [82]. An ensemble consists of at least two independent models, often called base model estimators, working in sync as a single model to predict the outcome of a process. Ensembles have the advantage of improving model accuracy and reducing generalization errors. An ensemble learning framework of ANNs, ANFISs, and support vector regressions (SVRs) was used to predict fifteen process parameters with at least 5% improvement when compared only with individual base models [59].

There are at least three basic ensemble learning methods as follows: simple averaging ensemble (SAE), a weighted average ensemble (WAE), and a nonlinear neural ensemble (NNE). In a study on the WWTP of Nicosia, Cyprus, an ensemble of feedforward neural networks (FFNNs), ANFISs, support vector machines (SVMs), and a classical multi-linear regression (MLR) were used for the prediction effluent BOD [83]. The study utilized and compared the three different ensemble methods, and the results showed that the ensemble models of the SAE, WAE, and NNE increased performance efficiency up to 14%, 20%, and 24%, respectively, in the prediction of effluent BOD. Table 4 provides a list of some ML-DM techniques used for WWTP control.

**Table 4.** Machine learning and data mining (ML-DM) AI techniques used for WWTP control.

| Model Type | Type of Test | Optimized or Predicted Variable | Metric of Evaluation | Reference |
|---|---|---|---|---|
| Multi-adaptive regression spline (MARS) | Offline modelling and optimization | *DO* | 31%+ reduction in airflow rate | [77] |
| Constrained Markov decision process (CMDP) | Offline modelling and optimization with field pilot test implementation | *DO*, WAS pump rate, and internal recycle pump rate | 13.5% energy reduction 14% less chemicals use for phosphorus 17% reduction in sludge production | [78] |
| Reinforcement learning (RL) | Simulation—BSM1 | N-ammonia | Cost reduction of N-ammonia removal | [80] |
| Direct heuristic dynamic programming (dHDP)-based RL. | Simulation—BSM1 | *DO* NO$_2$ | Single objective *DO* control design: IAE of 0.068 ISE of 0.00063 | [79] |
| Ensemble of feedforward neural network (FFNN), ANFIS, SVM, and a multi-linear regression (MLR) | Effluent quality parameter prediction | BOD | Comparison of ensemble techniques in terms of performance efficiency: SAE 14% WAE 20% NNE 24% | [84] |
| Ensemble of AdaBoost, gradient boost, and random forest regression | Effluent quality parameter prediction | TDS BOD5 COD | Adaboost TDS correlation coefficient = 0.96 Gradient boost BOD5 correlation coefficient = 0.90 COD correlation coefficient = 0.75 | [83] |

As established in earlier sections, an ensemble comprises at least two independent models working together through ensemble techniques such as SAE, WAE, or NNE. However, there are stand-alone ML-DM algorithms based on advanced ensemble techniques. Algorithms such as AdaBoost, extreme gradient boost (XGB), CatBoost, and RF are based on advanced bagging and boosting ensemble techniques. These algorithms can also be referred to as ensemble ML algorithms. They work with similar principles in that a subset of evenly weighted data is created from an original or universal dataset. An initial base model is created from that subset to predict the universal dataset. Then, the incorrectly predicted data points of the universal datasets are given higher weights, and another base model is created that is usually an improvement of the previous. The sequence continues until a final and optimum model, a weighted mean of all previous base models, is obtained.

The effluent quality parameters of a WWTP in Qom province, Iran, were predicted using AdaBoost, gradient boost, and random forest [83]. The results from this study showed that AdaBoost and gradient boost performed better at predicting total dissolved solids (TDSs) and $BOD_5$, respectively.

### 3.4.6. Real WWTP Implementation of AI Control

Publications on AI integration or experimentation within real WWTPs are limited. Typically, the assessment of AI control strategies is confined to dynamic simulation environments, rarely extending to real-world plant scenarios. Detailed case studies focusing on individual plants are less common compared to broader studies of multiple plants and their utilization of artificial intelligence. Most successfully implemented AI applications in real-world WWTPs are often marketed as software-as-a-service (SaaS) solutions. These successful applications often have a scarcity of comprehensive case studies and white papers detailing their reproducibility. Some successful implementations are mentioned below.

Aveva™, an esteemed industrial software solution provider headquartered in Cambridge, UK and recognized for its PI system product used in WWTPS, introduced a predictive analytics product [85]. This product boasts no-code AI and ML capabilities and has been asserted as a proven solution implemented by industrial operations for over 15 years. At the time of this review, no case studies or white papers had been made publicly available on the predictive analytics product.

Xylem Solutions, a water technology provider based in Washington, DC, USA, deployed a proprietary AI tool to optimize the efficiency of energy and chemical usage within the aeration process of a WWTP situated in Cuxhaven, Germany. The plant, which is operated by EWE WASSER GmbH (EWE) also situated in Cuxhaven, Germany, has a treatment capacity of 400,000 PE. According to reports, this implementation resulted in a remarkable 30% reduction in energy consumption while maintaining full compliance with regulatory standards [82]. The optimization tool in this case study, known as the Xylem Vue, uses neural network models of carbon, nitrogen, and phosphorus elimination processes. These models were developed using data extracted from the plant's SCADA system, allowing for a comprehensive data-driven approach to enhancing operational efficiency. The report in this case study showed that since its integration in 2017, the treatment plant achieved an annual saving of 1.2 million kilowatt-hours of aeration energy usage. This saving equates to powering 321 homes using 3500 kWh per year, showcasing the significant strides made in energy conservation [86].

In a separate case study, an advanced supervisory control system, known as PreviSys, was deployed to oversee operations at the Klimzowiec WWTP in Chorzow, southern Poland. PreviSys, a data-driven tool utilizing algorithms like model predictive control (MPC), was integrated into the plant's SCADA system. Its primary objective was to execute different operational strategies for optimizing the biological nutrient removal (BNR) process and enhancing the plant's energy balance [87]. The objective was realized through a comparative assessment between the default control strategy and the strategies provided by PreviSys, which compensated for uncertainties as a support operating system, ultimately leading to at least a 16% reduction in energy consumption at the Klimzowiec WWTP. Although AI

did not exclusively drive the MPC in this specific case study, it is noteworthy that MPC can be harmoniously fused with AI methodologies in specific applications.

In the area of monitoring WWTP performance, there is an increasing need to closely track treatment quality, especially with rising effluent quality standards and strict discharge regulations. A unique AI development in this field is a soft sensor powered by a fuzzy-neural network (FNN) [88]. This innovative approach was implemented in a study across two WWTPs situated in Beijing, China, with treatment capacities of 400,000 $m^3$/d and 1,000,000 $m^3$/d. Unlike traditional physical sensors which measure variables using physical hardware, soft sensors use algorithms to infer the variable based on other data. In this study, the soft sensor specifically monitored effluent concentrations of total phosphorus and ammonia nitrogen in two WWTPs. The soft sensor operated on an FNN that underwent training using an adaptive second-order algorithm. Its inputs were chosen through principal component analysis (PCA). After being put into production for a year at the two treatment plants, these soft sensors consistently achieved a quarterly mean accuracy of at least 90% [88].

### 3.4.7. Potential Disadvantages of Implementing AI Control in WWTPs

While the rapid evolution and promise of AI technology signify remarkable advancements, there exist specific concerns, challenges, and potential disadvantages that warrant attention within the wastewater treatment industry that encompass the following points:

Cost of implementation: the integration of AI systems typically demands substantial upfront investments in technology, specialized expertise, training, and infrastructure. While these costs are relatively quantifiable, there exists a less conspicuous yet critical factor—the cost of false positives [89]. For a WWTP, a false positive from an AI system could trigger violations of treatment and discharge regulations, setting off a cascade of environmental challenges. The repercussions might extend beyond immediate compliance issues, potentially leading to significant environmental consequences.

Dependency and adaptability: the concern regarding the adaptability of the workforce considering AI advancements is pervasive, particularly regarding potential job displacement [90,91]. Overdependence on AI systems may inadvertently erode human expertise and decision-making abilities, impacting the readiness of handling unforeseen situations.

Data quality: the effectiveness of AI strategies hinges significantly on the quality of input data, directly influencing the precision of output results. Any compromise in data quality poses a substantial risk to the overall efficiency of the strategy [89]. Factors like sensor malfunctions, breakdowns, or severe process variability can impact data quality. Process variability introduces the concern of concept drift, where AI-driven models or control strategies struggle to generate accurate outputs when faced with unfamiliar data instances. Therefore, addressing possible sensor breakdowns becomes a pivotal consideration for ensuring the long-term viability of AI strategies, potentially necessitating additional capital expenditure. Likewise, mitigating concept drift demands periodic model retraining, which may entail supplementary computing costs.

Data security and privacy risks: with increased connectivity, data usage, and the usage of AI tools, the risk of unauthorized access or sensitive information and cyber attacks increasingly becomes a sensitive issue. The implementation of AI strategies therefore requires a thorough evaluation of data and network security policies [89,92].

Ethics and AI bias: the ethical factors of AI deployment raise significant concerns, particularly regarding algorithmic transparency, biases, and accountability issues [89,93]. Reports have highlighted instances where AI tools lack transparency, potentially introducing biases into decision-making processes. Among these concerns is the one regarding how ML systems inherently exhibit biases due to their design from the collection of data and the subsequent preprocessing and feature engineering that aim to render the data more suitable for modeling purposes [94]. Several fundamental principles of ethics within AI have been identified including transparency, justice, non-maleficence, responsibility, privacy, trust,

sustainability, dignity, and solidarity [89,93–95]. These principles serve as crucial guidelines to be incorporated into the standards governing AI development and deployment.

## 4. Discussion and Research Gaps

From all the reviewed papers, data-driven control strategies have shown high accuracy and efficiency in the control of nonlinear processes. The application of data-driven strategies in the study on wastewater treatment has yielded profound results in various metrics such as energy consumption, treatment quality, airflow rate reduction, and operating cost index. It is noteworthy that within the previous decade, there has been a significant influx of over 20,000 publications and references to artificial intelligence (AI) and data-driven control strategies. However, upon conducting a comprehensive review of a representative sample from this vast body of work, it becomes apparent that less than 20% of these publications have been dedicated to the practical implementation of these strategies in full-scale or pilot-scale treatment plants. Most of these publications primarily revolve around simulation-based studies.

An evident trade-off can be observed between the simplicity of a data-driven strategy, the evaluation metric utilized, and the level of difficulty associated with its implementation. Typically, a more complex model tends to yield higher evaluation metrics but also presents challenges when it comes to implementing the strategy at a full-scale or pilot plant level. As a general guideline, an ideal data-driven control strategy aims to strike a balance of being sufficiently parsimonious in order to yield significant improvements in metrics while remaining simple enough to facilitate straightforward implementation within a full-scale or pilot plant setting.

As an example, when considering the implementation of a data-driven control strategy like reinforcement learning (RL), RL agents navigate towards an optimal control strategy by interacting with the environment based on rewards and penalties. This iterative process may continue until the RL agent has garnered rewards across a diverse range of plant disturbances, eventually achieving optimal control. However, it is important to acknowledge that during the time it takes for the RL agent to reach optimal control, the treatment plant may have already violated certain regulations. Consequently, it is understandable that no operator would willingly entrust the operation of a treatment facility solely to an RL agent or other unsupervised data-driven control strategies.

In practice, most operators would likely prefer to implement a precise data-driven control strategy that has been thoroughly validated to meet the operational requirements of another plant over a significant period, encompassing annual changes in seasons and various external disturbances beyond the scope of a simulation environment. However, a notable challenge arises with this preferred approach as data-driven control strategies are highly specific to individual processes or plants. Although there are some solutions available for pre-instructing an RL agent to bridge this implementation gap in wastewater treatment plants (WWTPs), these solutions remain experimental with anticipated improvements in the future [96].

An additional crucial yet often disregarded factor contributing to the limited success of translating AI research into full-scale implementation is the inadequate representation of domain knowledge and the involvement of diverse expert groups throughout the various stages of an AI project. Ensuring an equitable representation of domain expertise helps us to ensure the development of appropriate models that are tailored to meet the anticipated objectives and align with the existing infrastructure, thus enabling seamless integration. This collaborative approach enables the development of AI models that are well suited to address the intricacies of the target operations, enhancing the prospects of successful implementation.

From the representative body of the literature reviewed in this paper, FL and MPCs were observed as the most implemented AI techniques in either full or pilot-scale plants. This observation perhaps speaks of the trade-off between the complexity and difficulty of implementing AI control strategies. It was also observed that the common approach

towards the minimization of energy and the cost of operating a WWTP is the optimization of *DO*, but the theoretically recommended *DO* setpoint range is 2.0 ± 0.5 mg/L, which does not allow for a wide optimization search space. Another minimization approach would be to have a single objective optimization of the air flow rate with constraints on the final effluent properties to ensure that treatment quality is not compromised or to have a multi-objective optimization of the air flow rate and residual *DO* concentrations using desired final effluent characteristics as a constraint. A multi-objective optimization approach gives an operator a range of equally optimal solutions, while a single-objective optimization provides just one optimum point.

## 5. Conclusions

The water–energy nexus is a critical aspect of sustainable development, and wastewater treatment plays a crucial role in ensuring public health and safety. However, meeting regulatory standards often results in higher operating costs and energy requirements for wastewater treatment plants, which can be a major challenge for municipalities. The application of AI in controlling the cost and the energy-intensive processes of WWTPs has demonstrated significant potential for addressing this challenge. AI strategies with high accuracies capture nonlinearities in the system that are not accounted for when using mechanistic models. While most AI control strategies have not yet been widely implemented beyond scientific experimentation, some, such as FL and MPCs, are gradually being accepted into pilot and full-scale use. This transition from conventional controllers to AI strategies is likely to follow a similar path as the transition from on–off controllers to PIDs and cascade controllers. However, it is important to note that the implementation of AI strategies in wastewater treatment should be carried out in a parsimonious and responsible manner to ensure their effectiveness and sustainability.

Overall, the introduction of disruptive technologies into workspaces invariably faces skepticism and reluctance, and the integration of AI control within the WWT industry is no exception. The author's perspective on the evolution of AI in WWTP process control suggests an initial implementation in treatment units associated with minimal risks. Establishing trust in AI systems within these low-risk units could catalyze their wider adoption in more complex and higher-risk units. Consequently, this progression will necessitate the development and implementation of robust policies and guidelines governing the utilization of AI across varying levels of risk. Establishing these frameworks becomes pivotal in ensuring the responsible and effective deployment of AI within the wastewater treatment industry.

**Author Contributions:** Conceptualization, C.M. and M.S.Z.; methodology, C.M. and M.S.Z.; formal analysis, M.S.Z.; investigation, C.M.; resources, G.A.; data curation, C.M.; writing—original draft preparation, C.M.; writing—review and editing, C.M., M.S.Z., D.K. and G.A.; visualization, C.M.; supervision, D.K. and G.A.; project administration, D.K. and G.A.; funding acquisition, D.K. and G.A. All authors have read and agreed to the published version of the manuscript.

**Funding:** This research was funded by Natural Sciences and Engineering Research Council of Canada (NSERC) and The City of Calgary with grant numbers 10035300 and 10030193 respectively. The APC was funded by NSERC and The City of Calgary.

**Acknowledgments:** The authors would like to acknowledge the support of the Natural Sciences and Engineering Council of Canada and the City of Calgary.

**Conflicts of Interest:** The authors declare no conflict of interest.

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
