# Peer review of "A Review of AI-Driven Control Strategies in the Activated Sludge Process with Emphasis on Aeration Control"

_water, doi:10.3390/w16020305_

Round 1

Reviewer 1 Report

Comments and Suggestions for Authors

Dear Authors,

I have thoroughly enjoyed reading your article, particularly as it aligns closely with my field of work. I would like to recommend the inclusion of the following aspects to enrich the content:

Incorporate real-world examples of industrial WWTPs where AI has been successfully applied in process control.

Explore the potential disadvantages of implementing this technology, such as increased monitoring, the need for additional analysis instruments, heightened maintenance requirements, and associated costs.

Please provide your insights into how you envision future WWTPs evolving with the implementation of AI in process control.

These additions would further enhance the depth and relevance of your article.

Comments on the Quality of English Language

English language is fine. 

Reviewer 2 Report

Comments and Suggestions for Authors

The review article A Review of AI-Driven Control Strategies in the Activated Sludge Process with Emphasis on Aeration Control presented by the is valuable and suitable for publishing in Water. The article is very interesting and important for scientists associated in technology for control and optimizing wastewater treatment to reduce energy consumption and high efficiency of WWTPs operation.

I believe my detailed comments will be useful.

Abstract

In my opinion your abstract is too general.

You can add some information from chapter 3.5.1

Abstract is very important part of review paper.

Also check please where is chapter 3.5? I could not find it.

Introduction-

In the end you wrote about paper organization. Add short message to riders what was the practical and scientific aim of your review.

Energy consumption during wastewater treatment is not only aeration-there are many other processes including, for example, aerobic or anaerobic sewage sludge treatment and the problem of process water after such a treatment that generate energy consumption in aeration chambers.

In this chapter you are writing mainly about energy consumption. Please write more about AI in WWTP operation (possible in the future probably).

Also analyzing energy consumption one more indicator is very important- energy consumption related to the pollutant load removed

Chapter 2 presents well-known information - this chapter can be reduced

Line 226

Please check titles and numbering of your chapters. It should be 2.2.2? or 3 Aeration control

Line 311

Removal of nitrogen instead “Removal of ammonia and nitrogen”

Line 315

This is an article not a textbook for students- remove such information as equation 8.

What was the reason to present figures 6 and 7? The growing number of publications on mechanistic and AI models?

Figure 8 is important please describe it more in this chapter.

Line 392

Table 1- check the title please

Some information from chapter 4 can be moved to introduction-e.g. line 578-588

Also chapter Conclusion can be improved- it is too general in my opinion.
